# A Nano Refractive Index Sensing Structure for Monitoring Hemoglobin Concentration in Human Body

**DOI:** 10.3390/nano12213784

**Published:** 2022-10-27

**Authors:** Guoquan Zhou, Shubin Yan, Lili Chen, Xiaoyu Zhang, Lifang Shen, Pengwei Liu, Yang Cui, Jilai Liu, Tingsong Li, Yifeng Ren

**Affiliations:** 1School of Electrical and Control Engineering, North University of China, Taiyuan 030051, China; 2School of Electrical Engineering, Zhejiang University of Water Resources and Electric Power, Hangzhou 310018, China; 3Joint Laboratory of Intelligent Equipment and System for Water Conservancy and Hydropower Safety Monitoring of Zhejiang Province and Belarus, Hangzhou 310018, China

**Keywords:** nanosensor, Fano resonance, electrolyte concentration measurement, detect the concentration of hemoglobin in blood

## Abstract

This paper proposes a nanosensor structure consisting of a metal–insulator–metal (MIM) waveguide with a rectangular root and a double-ring (SRRDR) with a rectangular cavity. In this paper, the cause and internal mechanism of Fano resonance are investigated by the finite element method (FEM), and the transport characteristics are optimized by changing various parameters of the structure. The results show that the structure can achieve double Fano resonance. Due to the destructive disturbance between the wideband mode of the inverted rectangle on the bus waveguide and the narrowband mode of the SRRDR, the output spectrum of the system shows an obvious asymmetric Fano diagram, and the structural parameters of the sensor have a great influence on the Fano resonance. By changing the sensitive parameters, the optimal sensitivity of the refractive index nanosensor is 2280 nm/RIU, and the coefficient of excellence (FOM) is 76.7. In addition, the proposed high-sensitivity nanosensor will be used to detect hemoglobin concentration in blood, which has positive applications for biosensors and has great potential for future nanosensing and optical integration systems.

## 1. Introduction

**Surface plasmon polaritons** (SPPs) are electromagnetic waves that can only propagate at the dielectric metal interface [1,2]. They are formed by the interference of free electrons and incident photons of metal [3,4]. The electromagnetic wave can be transmitted on the interface, and most of the field energy is limited on the interface, so its intensity decreases rapidly in the direction perpendicular to the interface [5,6]. SPPs are generated when the horizontally polarized wave is incident on the interface between a metal and a medium at a certain angle [7,8]. The incident electric field will cause the charge change at the interface between metal and medium, and the charge will undergo collective oscillation [9,10]. At the same time, the radiation field of the oscillating charge also permeates into the metal, and the local electric field increases at the interface and decays exponentially along the normal direction of the interface [11,12].

The mode of SPP can be divided into two categories according to the difference of the excited metal structure [13,14]: 1. local surface plasmon polaritons bound to the tip of a needle or particles, and 2. surface plasmon polaritons propagating on the surface of metal and medium [15,16]. SPPs break the limits of conventional optical diffraction, and the structure of propagating surface plasmon polaritons (SPPs) is larger than the wavelength of incident light. SPPs are highly sensitive to changes in nanostructures and media types [17,18]. We usually adapt the structure dimensions to change the shape of the resonance curve to optimize the sensing performance of the device. The most prominent feature of Fano resonance is that it can have asymmetric sharp spectral lines [19,20]. In recent years, the Fano resonance experimental research on MIM waveguide structure has been increasing. Chen et al. designed a refractive index sensor with a sensitivity of 1120 nm/RIU [21], and Zhang et al. proposed a refractive index nano sensor with a sensitivity of 1268 nm/RIU [22]. Wang et al. proposed a T-shaped cavity with a sensitivity of 680 nm/RIU [23].

In this paper, a simple structure consisting of an anti-rectangular MIM waveguide and a double-ring rectangular resonator (SRRDR) is presented and investigated. A finite element method (FEM) is introduced to analyze the transmission spectrum and normalized magnetic field distribution. Compared with the single-ring structure, the disk-shaped cavity with a double-ring structure has more advantages in capturing light. Moreover, the double-ring structure can obtain two clear Fano resonance curves, and the double-ring structure increases the detection range of the sensor. We not only study the relationship of Fano resonance to shift with the change of refractive index but also dissect the effect of geometrical parameter changes on the resonance plot. Furthermore, the application of the designed structures in temperature sensing and biosensing structures is also investigated in detail. Taking hemoglobin as an example, its concentration can be detected, which provides a theoretical basis for nano biosensor equipment.

## 2. Structural Model and Analysis Method

In consideration of the fact that the structure designed is much thicker than the surface depth of the SPP, we used a 2D model instead of a 3D model for simple calculations [24,25]. Figure 1 shows the 2D schematic of the structure. Figure 2 shows a three-dimensional model. The whole sensor is symmetrical along the centerline, and it consists of a waveguide with an inverted rectangle and an SRRDR. R and r, respectively, represent the outer radius and inner radius of the ring, the height of the waveguide rectangle is represented by H, the center distance of the double ring is represented by s, the length and width of the rectangular rod on the double ring are represented by L and d, and the rotation angle of the rectangular rod on the double ring is represented by φ, where g denotes the gap between the inverted rectangular waveguide and the SRRDR, and ω represents the width of the waveguide. The color area is filled with silver as the filling medium, the white part is filled with air, and the material of the base is medium quartz. The advantage of choosing silver as the filler metal is that it can ensure the high field of the system and reduce the power consumption. Regarding the fabrication of silver layers, we can prepare enough silver layers on the substrate by vapor deposition (CVD), and the desired structure is obtained by an electron beam etching method. We can calculate the relative permittivity of silver through the dispersion model [26,27] ($\varepsilon_{\mathrm{Air}}=1$).
(1)$$\varepsilon \left(\omega \right)=\varepsilon_{\infty }+\frac{\varepsilon_{s}-\varepsilon_{\infty }}{1+i\tau \omega }+\frac{\sigma }{{i\omega \varepsilon }_{0}}$$
where ω is the angular frequency of light, (=1.38 × 1016) can be expressed as the plasma frequency of silver, and τ (τ = 7.35 × 1015) represents the average free time [28].

TM mode equation is:(2)$$\tanh\left(k\omega \right)=-\frac{2{k\alpha }_{c}}{k^{2}+p^{2}\alpha_{c}}$$
where k is the wave vector, $\alpha_{c}$ = ${\left[k_{0}{}^{2}\times \left(\varepsilon_{\mathrm{in}}-\varepsilon_{m}\right)+k\right]}^{1/2}$, and $k_{0}$ = $2\pi /\lambda_{0}$. Here, $P=\varepsilon_{\mathrm{in}}/\varepsilon_{m}$, $\varepsilon_{\mathrm{in}}$, $\varepsilon_{\mathrm{out}}$ dielectric constant from edge insulator to point metal, respectively [29].
(3)λm=2ReneffLm−Ψrπm=1,2,…,
(4)Re(neff)=εm+k/k02
where L represents the circumference of the ring cavity, ${\;\Psi }_{r}$ denotes the phase shift due to the SPP reflection from the boundary surface of the metal insulator, and m is the positive integral, namely the reverse pole number of the SPP. The characteristics of the sensor can be evaluated by two important parameters: sensitivity (S) and FOM, which are expressed in the following equation [30,31]:(5)S=Δλ/Δn,
(6)FOM=S/FWHM,
where n is the changes in the resonance wavelength and the refractive index, respectively [32].

Considering the manufacturing cost of nanoscale devices, Pre-simulation allows tuning the mechanisms of the sensor structure and optimizing the performance of the nanosensor. In addition, the two-dimensional model structure of the SRRDR is constructed by COMSOL 5.4, and the structure is simulated and analyzed. In order to ensure the reliability of the calculation results, our grid is divided by hyper refinement. For the selection of light source, we finally use laser. The laser has good monochromatic aberration and correlation. A well-matched boundary is applied as a layer to absorb the spilled light waves.

## 3. Results and Discussion

In order to better illustrate the transmission characteristics of the designed structure, we compare the whole system, single band reverse rectangular waveguide and the SRRDR structure. The reverse rectangular waveguide structure and the SRRDR structure are depicted in Figure 3 and Figure 4, respectively. The transmission spectrum of the whole system is shown in Figure 5. The green, blue, and red continuous lines shown represent the transmission spectra of the whole system, the single SRRDR structure, and the inverted rectangular waveguide structure, respectively. From the transmission plot, we can obtain the spectral information of a single SRRDR structure, with low transmittance and relatively symmetrical shape. This is similar to Lorentz resonance, which can be regarded as a narrowband mode directly excited by incident light, and the transmittance is low, so it is regarded as a narrowband mode. The entire curve of the transmission spectrum of a single inverted waveguide structure has a positive slope and has a high transmittance. Therefore, it is considered a continuous wideband mode. In the complete structure, we can see an obvious asymmetric curve, which indicates that the structure has a good Fano resonance phenomenon. The reason for this phenomenon is caused by the interaction between the continuous wideband state generated by the rectangular waveguide and the discrete narrowband state of a single SRRDR. Next, we will discuss the influence of different parameters on the transmission spectrum.

The default parameters are as follows: R = 230 nm, r = 180 nm, w = 50 nm, L = 220 nm, d = 50 nm, g = 10 nm, φ = 30°, and h = 90 nm. (This is the optimal sensitivity structure).

Initially, we studied the refractive index. We set up six different refractive data for experiments. The resonance diagram is shown in the figure, and the simulation results are shown in Figure 6a,b. With the increase of the equal distance of n, the transmission spectrum will also be an approximately equidistant red shift. We also performed linear fitting for the change of inclination wavelength. The optimal sensitivity of the sensor is 2280 nm/RIU, and the advantage value is 76.7.

Subsequently, we study whether the Fano resonance curve will have different changes when the radius of the SRRDR is selected. The radius R1 varies from 190 nm to 230 nm. The transmission spectrum and sensitivity curve of the SRRDR are shown in Figure 7a. The most obvious one is that with the continuous increase of radius, the transmission spectrum will undergo continuous red shift, and the transmittance and FWHM will fluctuate slightly. In addition, we also perform linear fitting according to the resonance curves of different radii. As shown in Figure 7b, the radius of the coupling ring keeps increasing, and the sensitivity of the sensor also increases. This shows that R1 can have a great influence on the tilt wavelength. When R1 = 230 nm, the maximum sensitivity is 2280 nm/RIU. FOM was 76.7.

In order to further explore the factors affecting the accuracy of the sensor, we also carried out experiments on the rectangular cavity on the double ring. The resonance curve changes as the length and width of the rectangular cavity keep changing. As illustrated in Figure 8, we increased the length of the rectangular cavity from 180 nm to 220 nm and the width from 30 nm to 50 nm. During this process, with the continuous change of the length and width of the rectangular cavity, the Fano curve showed an obvious red shift, and the figure basically did not change. The reason is that the SPP can be better excited by proper adjustment of the rectangular cavity. According to the fitting straight line in Figure 9, the length and width of the rectangular cavity have a great influence on the accuracy of the sensor. The sensitivity is from 1990 nm/RIU to 2280 nm/RIU, the maximum advantage value is 76.7, l = 220, d = 50 nm, which is the best choice for structure and performance. 

In order to further study the effect of the coupling gap. We superposed the coupling gap layer by layer from 10 nm to 30 nm. The research coupling results of the SRRDR and waveguide straight line are shown in the Figure 10a. The figure shows that with the continuous increase of g, the transmittance of the most obvious Fano resonance curve is increasing, which indicates that the coupling performance is greatly weakened. This will inevitably lead to a sharp deterioration in the performance of the sensor. In addition, the Fano curve is blue shifted. FWHM also tends to narrow. The FWHM diagram is shown in Figure 10b.

It is obvious that the sensitivity of the sensor decreases sharply with the increase of g. The coupling results also appear unsatisfactory with the increase of g. After comparison, g = 10 nm is the best parameter in this design. When the sensitivity is 2280 nm/RIU and FOM is 76.7, the best performance parameter can be obtained.

Finally, we studied the influence of the height of the reverse waveguide on the resonance curve. We gradually increased h from 50 nm to 90 nm. The simulation results are shown in Figure 11a. We can see that no matter how the height of the reverse rectangle changes, the inclination wavelength of the Fano resonance line and the projection rate has hardly changed. On the contrary, the reverse waveguide can enhance the coupling effect between SPP and the SRRDR. After adding the reverse waveguide, the Fano resonance curve becomes more obvious. FWHM has great influence on the monochromatic property of the light wave, and the proper length of the rectangular waveguide also has some influence on FWHM. Through experimental comparison, when h = 90 nm, FWHM becomes the smallest, 29 nm. The change diagram of FWHM is shown in Figure 11b.

The sensitivity and merit values of the structure designed in this paper are significantly better than the mentioned structures, and the comparison parameters are shown in Table 1.

## 4. Application

In this paper, a new type of nano biosensor is designed, which not only solves the problem that traditional optical devices cannot be integrated on a large scale due to diffraction limit, but also meets the challenge of designing a double-loop structure that can output two different Fano resonance curves, which can not only perform multi-frequency regulation, but also greatly increase the detection range of the sensor [33,34]. We filled the waveguide and double-ring resonator in the design structure with appropriate blood samples. From the data, a positive linear relationship between hemoglobin concentration and refractive index was observed, and we obtained the concentration of hemoglobin in the blood to be measured (the experiment measured the hemoglobin concentration in dried blood). The function correspondence is as follows [35,36,37]:(7)$$n=n_{0}+\alpha C+\beta T+\gamma_{1}\lambda +\gamma_{2}\lambda^{2}+\gamma_{3}\lambda^{3},$$

In the above functional equation, $n_{0}\;$= 1.54712, where C  means the dry hemoglobin concentration content (g/L), and α represents the specific refractive index increment. The data are illustrated in Table 2. Due to the different hemoglobin concentrations of different blood types in the human body, the hemoglobin values will also change. The hemoglobin values of type A blood, type B blood, and type O blood are 0.0009014, 0.001109, and 0.001126, respectively [38,39]. Figure 12 shows data for different blood types.

For the sensitivity of the structure to measure hemoglobin, the structure parameters were set as follows: r = 230 nm, r = 180 nm, l = 220 nm, and d = 50 nm. At this time, the structural sensitivity is 2280 nm/RIU. Considering the hemoglobin concentration range of the normal human body, the collection range is 108.9–146.2 g/L, which meets the working range of the sensor. With the increase of hemoglobin concentration of different blood types, the projection spectra of type A, type B, and type O blood have changed, and the Fano resonance curve has a red shift. This shows that even if the hemoglobin concentration changes significantly, the sensing data is still reliable, and due to the resonance splitting, the sensing structure can measure different concentrations of hemoglobin at the same time, which has a wider application.

Due to 0.01 nm spectrum, displacement is easy to detect, so high sensitivity of the proposed structure can be easily detected by different blood types. It can detect the concentration of hemoglobin, providing a theoretical basis for nano biosensor devices. The designed sensor is not only small in size, high in accuracy, and easy to integrate, but also has great development space in the future nano sensing and optical communication system.

The analyzed sensing structure can also be used for sensing in aqueous solutions. The structure can be used as ethanol for temperature sensing. Figure 13 shows the sensing characteristic curve. The expression of the refractive index with ambient temperature is as follows.
(8)$$n=1.36048-3.94\times 10^{-4}\times \left(T-T_{0}\right)$$

T denotes the temperature at which we conducted the experiment, and T_0_ is our constant temperature of 20 °C.

Considering that the boiling point of alcohol is about 78 °C and the temperature sensing data does not exceed 70 °C, Figure 13a shows that the alcohol temperature sensing also has a good sensing effect, and the resonance tilt wavelength will be blue-shifted as the temperature keeps rising. Moreover, the refractive index has a good linear relationship with temperature as shown in Figure 13b, and the accuracy of the sensor reaches 2500 nm/RIU. This demonstrates that the ethanol temperature-sensing structure has a good development space.

Compared with traditional glucose concentration detection, the design of this paper not only has a high figure of merit (FOM), but also can measure different concentrations of hemoglobin in this structure compared with the glucose concentration structure that can only perform a single measurement. By simulating the structural parameters that affect performance, this design can be used to measure different concentrations of hemoglobin simultaneously by changing the structural parameters. This greatly improves the efficiency of the measurement, which is far from being achieved by a single glucose measurement sensing structure.

## 5. Conclusions

In this paper, a plasma structure consisting of a rectangular MIM waveguide and a connected double-ring resonator (SRRDR) is proposed and studied theoretically. Different from the traditional rectangular cavity structure, the double-ring cavity is used instead of the rectangular cavity to better capture light and improve the sensing performance, and the output multiple resonance curve can adjust the spectrum in multiple different frequency bands. The application range of the sensor is greatly enhanced. This paper not only analyzes the effect of refractive index change of the medium on the resonance transmission spectrum curve, but also investigates the effect of various structural parameters on the resonance transmission spectrum curve. The sensitivity of the proposed structure is significantly improved, reaching 2280 nm/RIU, and the FOM is also 78.6.

Furthermore, the structure can be applied in the biological field. For example, we can analyze the hemoglobin content in blood by taking advantage of the different hemoglobin concentration content of different blood types. Thus, it is possible to detect many indicators related to hemoglobin. The structure provides a theoretical basis for nanobiosensor devices. The designed sensor is not only small in size, high in accuracy, and easy to integrate, but also has great potential for future nano-sensing and optical communication systems.

## Figures and Tables

**Figure 1 nanomaterials-12-03784-f001:**
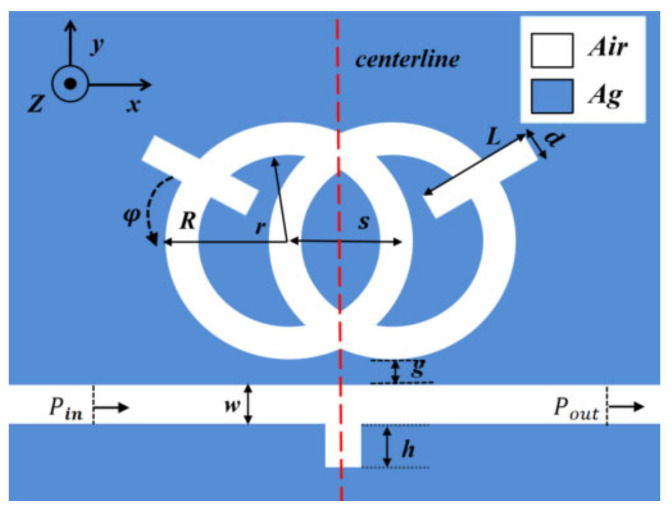
2D structural diagram.

**Figure 2 nanomaterials-12-03784-f002:**
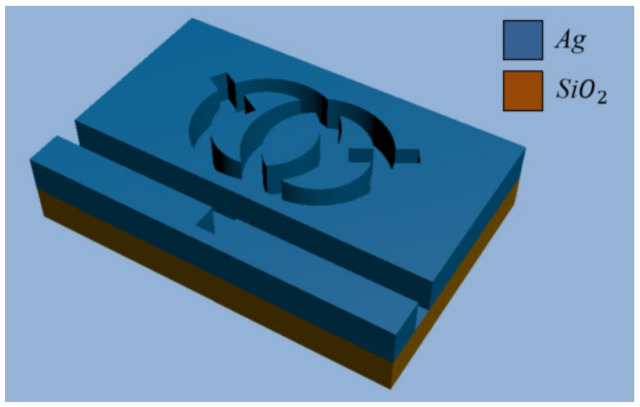
3D structural diagram.

**Figure 3 nanomaterials-12-03784-f003:**
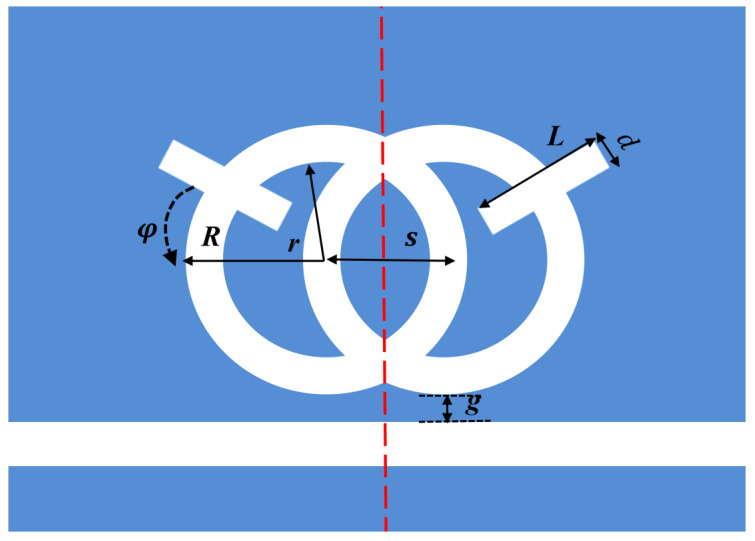
Single SRRDR.

**Figure 4 nanomaterials-12-03784-f004:**
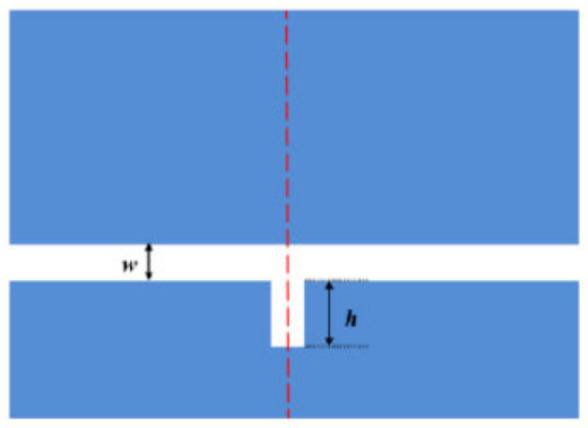
Single stub.

**Figure 5 nanomaterials-12-03784-f005:**
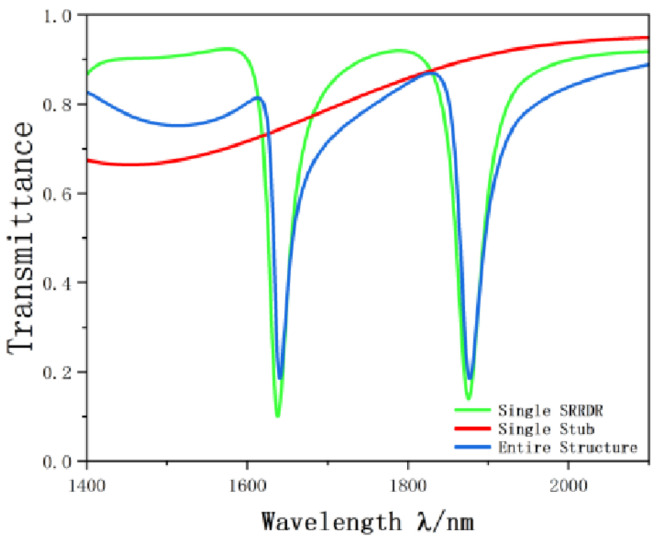
Schematic of projection spectrum.

**Figure 6 nanomaterials-12-03784-f006:**
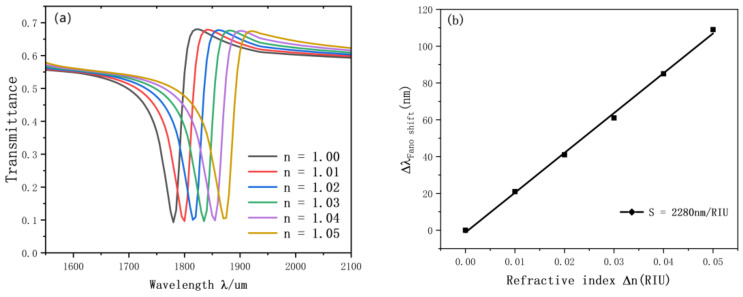
(**a**) Transmission spectrum of g variation; (**b**) the sensitivity of n changes to fit the line.

**Figure 7 nanomaterials-12-03784-f007:**
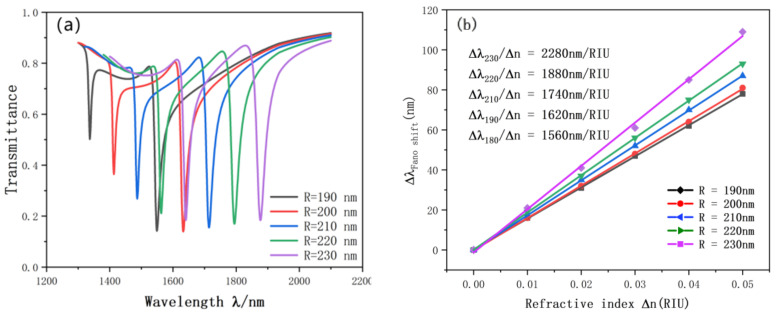
(**a**) Transmission spectrum of R variation; (**b**) the sensitivity of R changes to fit the line.

**Figure 8 nanomaterials-12-03784-f008:**
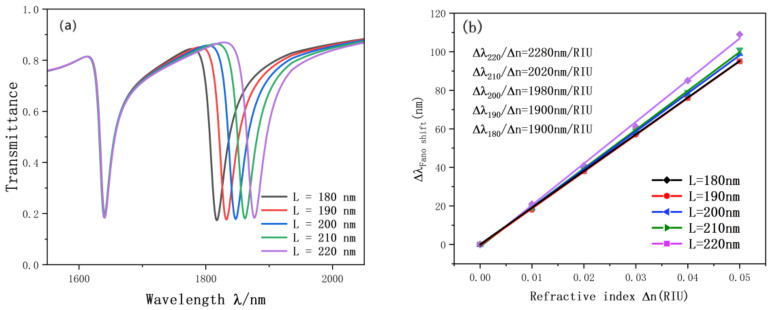
(**a**) Transmission spectrum of g variation; (**b**) the sensitivity of L changes to fit the line.

**Figure 9 nanomaterials-12-03784-f009:**
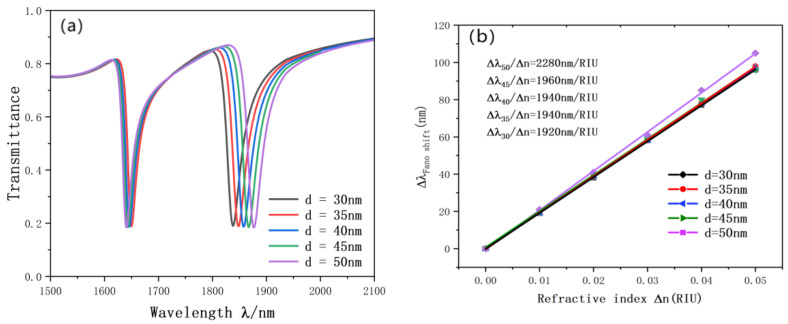
(**a**) Transmission spectrum of g variation; (**b**) the sensitivity of d changes to fit the line.

**Figure 10 nanomaterials-12-03784-f010:**
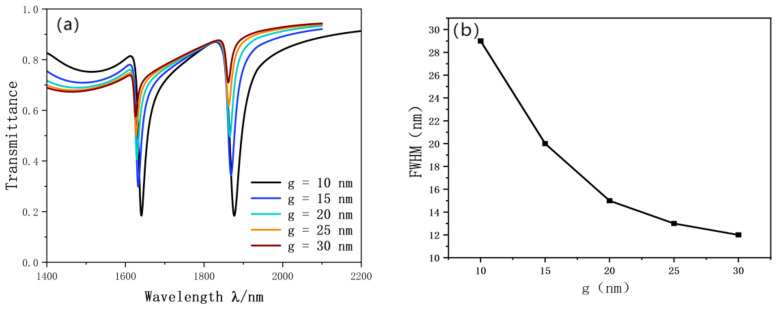
(**a**) Transmission spectrum of g variation; (**b**) trends in FWHM.

**Figure 11 nanomaterials-12-03784-f011:**
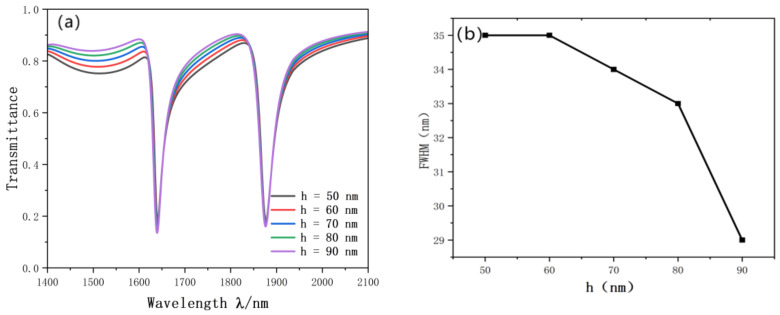
(**a**) Transmission spectrum of h variation; (**b**) trends in FWHM.

**Figure 12 nanomaterials-12-03784-f012:**
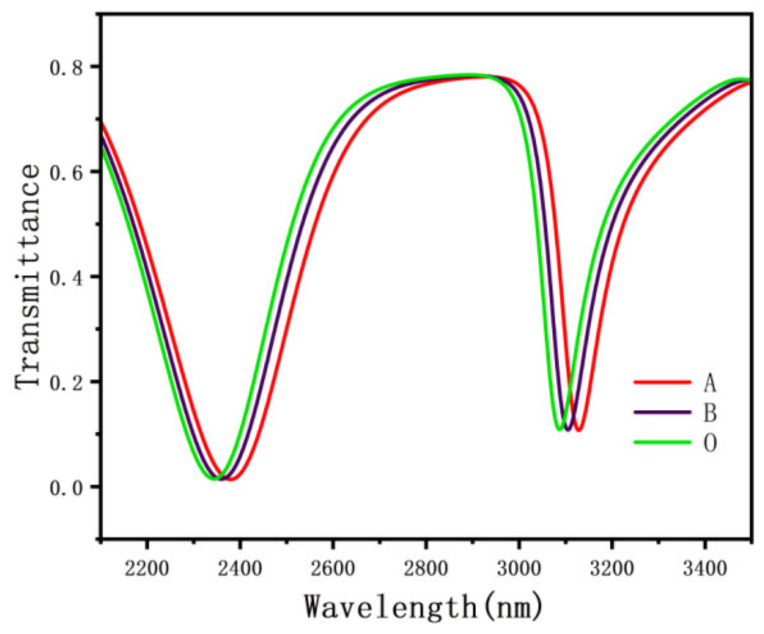
The transmission spectrum of different blood types.

**Figure 13 nanomaterials-12-03784-f013:**
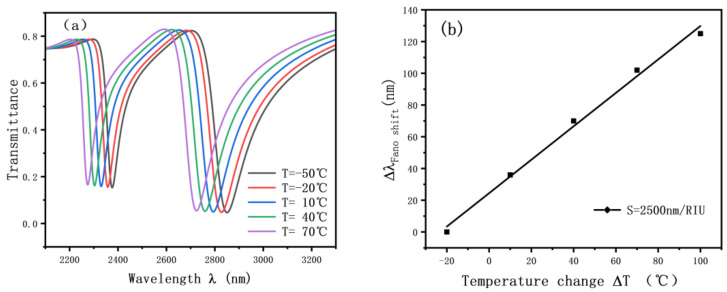
(**a**) Transmission spectrum of alcohol sensing; (**b**) fitting straight lines for alcohol sensing.

**Table 1 nanomaterials-12-03784-t001:** Comparison of structural data.

References	Structure Type	Sensitivity (nm/RIU)	Operating Wavelength
[17]	Circular split-ring resonator	1180	500–1200 nm
[18]	Cross-shaped cavity and baffle	1075	800–1300 nm
[19]	Isosceles triangular cavity	1200	1000–1400 nm
This work	The DRRR structure	2280	1450–2000 nm

**Table 2 nanomaterials-12-03784-t002:** Model coefficients.

Parameter	Unit	A-Group	B-Group	O-Group
$n_{0}$	-	1.54712	1.54712	1.54712
α	L/g	9.014 ×$\;10^{-4}$	11.09 × $10^{-4}$	11.26 × $10^{-4}$
∁	g/L	146.2	111.1	108.9
β	L/K	−6.497 × $10^{-5}$	−6.497 × $10^{-5}$	−6.497 × $10^{-5}$
$\gamma_{1}$	${\mathrm{nm}}^{-1}$	−8.47 × $10^{-6}$	−8.47 × $10^{-6}$	−8.47 × $10^{-6}$
$\gamma_{2}$	${\mathrm{nm}}^{-2}$	7.08 × $10^{-7}$	8.014 × $10^{-7}$	7.742 × $10^{-7}$
$\gamma_{3}$	${\mathrm{nm}}^{-3}$	−1.28 ×$\;10^{-10}$	−2.286 × $10^{-10}$	−1.823 × $10^{-10}$

## Data Availability

The data presented in this study are available on request from the corresponding author.

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
