# Peer review of "A Nano Refractive Index Sensing Structure for Monitoring Hemoglobin Concentration in Human Body"

_nanomaterials, 2022, doi:10.3390/nano12213784_

Round 1

Reviewer 1 Report

The manuscript entitled  “A nanoscale structure based on an MIM waveguide coupled with a double ring resonator for monitoring hemoglobin concentration in the human body” presents a theoretical analysis of spectral characteristics of a novel sensing structure based on a rectangular nano-waveguide coupled with a double –ring nanoresonator. Fano resonances of such a structure can be used for sensing. This manuscript is similar with previous author’ s manuscripts.  One can generally say, that theoretical analysis of novel sensing structures can highly improve the structure preparation as well as the interpretation of results and thus it is highly appreciated. Unfortunately, the presented manuscript is not very convincing in this respect and has to be improved. Particularly,

1.       Geometrical parameters of each analyzed structure have to be specified in order to a reader could make its own calculations. These specifications of parameters w,h,g R,r,s, L,d, j could be added in captions of Figs. 5-12.

2.       The analyzed sensing structure could be useful for sensing in aqueous solutions, i.e. for refractive-index values around 1.32. An analysis and optimization of a sensing structure for such solutions should be added to a novel manuscript.

3.       The analysis of the hemoglobin sensor is not clear. It is not specified which reference Eq. 7 comes from  because a reader can’ t find it neither in paper [37] nor [38]. Moreover, n0 for human blood has to be on a level 1.32 and not 1.54712. It is necessary to discuss ways for distinguishing different types of hemoglobin with the sensing structure if they present in a sample.

In addition, some technical improvements of the manuscript have to be made. Thus, it is strongly recommended not to use abbreviations such as MIM in the manuscript title. On the other hand, an abbreviation SPP in Introduction is not explained.

In Abstract, the last sentence related to the hemoglobin detection  is not very reliable in context with comment #3 above.

Formally, Eq. (5] does not correspond with S determined e.g. from Fig. 6b because there are changes of land n already used on the axes.

A sentence “In the above functional equation, =1.54712, where c means the hemoglobin concentration content (g/L), and represents the specific refractive index increment that varies with the hemoglobin concentration” on Page 7. Is confusing for a reader. Probably, n0 is missing. Furthermore, the increment a does not varies with the hemoglobin concentration. If there is a linear relation between the shift of resonance minimum and refractive index and the refractive index depends linearly on the concentration, there is no reason for last two sentences on Page 7. I recommend to discuss the sensor selectivity, instead.

The last paragraph of Conclusions is not supported by manuscript results.

I recommend to read carefully any novel manuscript to avoid some typing errors.

Reviewer 2 Report

I determined that this research is a validation study of reflective index-based optical (infrared) glucose sensor design using simulation (FEM). I think it is more appropriate for this paper to be published in Sensors than in Nanomaterials, but I think it is somewhat in line with the topic covered by Nanomaterials. However, the following matters must be faithfully supplemented before accepting publication.

(1) This study presents only simulation results, and it is necessary to explain in detail what differences may occur if used in actual experiments.

(2) In a section of Conclusion, authors briefly present the designed structure and sensor applications, but in order to explain the legitimacy of this study, detailed applications should be added with reference articles of related researches.

(3) When this sensor is attached on the human body and measures glucose concentrations, is there any advantage other than sensitivity compared to the structures presented as controls (Table 1)? The explanation for this issue should be supplemented.

(4) I think the addition of a description of each parameter in a caption of Figure 1 is needed to improve the reader's understanding.

Round 2

Reviewer 1 Report

I can accept most of changes made in the novel manuscript. Unfortunately, the section of the manuscript dealing with hemoglobin is still not correct. The reason is in Eq. (7) that does not correspond to refractive-index data reported ,e.g., in reference [35], Fig. 9. Values calculated from Eq. (7) are much higher than those shown in  Fig. 9 from [35]. Thus, the analysis of the hemoglobin sensor is not realistic and has to be improved to take into account experimental data. I can also recommend to check carefully the coefficients reported in Tab. 3 reference [35]. Furthermore, the increment a does not depend on the hemoglobin concentration but describes the dependence of the refractive index of blood on this concentration.  

Author Response

请参阅附件

Round 3

Reviewer 1 Report

I accept changes made in last manuscript and recommend it for the publication.